# Deep Supervised Summarization:
# Algorithm and Application to Learning Instructions

**Chengguang Xu**
Khoury College of Computer Sciences
Northeastern University
Boston, MA 02115
xu.cheng@husky.neu.edu

**Ehsan Elhamifar**
Khoury College of Computer Sciences
Northeastern University
Boston, MA 02115
eelhami@ccs.neu.edu

## Abstract

We address the problem of finding representative points of datasets by learning from multiple datasets and their ground-truth summaries. We develop a supervised subset selection framework, based on the facility location utility function, which learns to map datasets to their ground-truth representatives. To do so, we propose to learn representations of data so that the input of transformed data to the facility location recovers their ground-truth representatives. Given the NP-hardness of the utility function, we consider its convex relaxation based on sparse representation and investigate conditions under which the solution of the convex optimization recovers ground-truth representatives of each dataset. We design a loss function whose minimization over the parameters of the data representation network leads to satisfying the theoretical conditions, hence guaranteeing recovering ground-truth summaries. Given the non-convexity of the loss function, we develop an efficient learning scheme that alternates between representation learning by minimizing our proposed loss given the current assignments of points to ground-truth representatives and updating assignments given the current data representation. By experiments on the problem of learning key-steps (subactivities) of instructional videos, we show that our proposed framework improves the state-of-the-art supervised subset selection algorithms.

## 1   Introduction

Subset selection, which is the task of finding a small subset of most informative points from a large dataset, is a fundamental machine learning task with many applications, including, procedure learning [1, 2, 3], image, video, speech and document summarization [4, 5, 6, 7, 8, 9, 10, 11], data clustering [12, 13, 14, 15], feature and model selection [16, 17, 18, 19], social network marketing [20], product recommendation [21] and sensor placement [22, 23]. Subset selection involves design and optimization of utility functions that characterize the informativeness of selected data points, referred to as representatives. Different criteria have been studied in the literature, including (sequential) facility location [24, 2, 1] maximum cut [25, 26], maximum marginal relevance [27], sparse coding [28, 29] and DPPs [11, 30, 31]. Given that almost all subset selection criteria are, in general, non-convex and NP-hard, approximate methods, such as greedy algorithms for optimizing graph-cuts and (sequential) facility location [24, 32, 2], sampling from Determinantal Point Process (DPP) [11, 31] and convex relaxation-based methods [12, 33, 29, 34, 35, 36] have been studied in the literature.

Existing work on subset selection can be divided into the two main categories of unsupervised and supervised methods. The majority of existing research on subset selection falls into the unsupervised category, where one finds representatives of a dataset by optimizing the above criteria [5, 6, 7, 8, 9, 10, 11, 15, 22, 12, 28, 29] or others, such as diversity or coverage [37, 38, 39, 40, 41], importance

[42, 43, 5, 6, 44, 45, 46] and relevance [47, 39, 42, 48, 49]. The results are subsequently evaluated qualitatively or quantitatively against ground-truth representatives.

**Supervised Subset Selection.** Humans perform remarkably well in summarization of video and speech data, e.g., describe the content of a long complex video by a few sentences or by selecting a few frames/segments. This has motivated the development and study of supervised subset selection techniques that learn from human, with the goal of bringing high-level reasoning and incorporating user preferences into subset selection. More formally, in the supervised setting, given datasets and their ground-truth representatives, one tries to train subset selection to recover the ground-truth summary of each training dataset and to generalize to new datasets.

Despite its importance, supervised subset selection has only been more recently studied in the literature [8, 50, 51, 52, 53, 54, 55, 56, 30]. One difficulty is that supervised subset selection cannot be naively treated as classification, since, whether an item receives the label 'representative' or 'non-representative' depends on its relationships to the entire data. For example, a representative car image among images of cars, once considered in a dataset of face images will become non-representative. To address the problem, [50, 52] try to learn a combination of different criteria, i.e., weights of a mixture of submodular functions. However, deciding about which submodular functions and how many to combine is a non-trivial problem, which affects the performance. On the other hand, [8, 51, 53, 54, 30, 55] learn a DPP kernel or adapt it to test videos, by maximizing the likelihood of the ground-truth summary under the DPP kernel. However, maximizing the summary likelihood for the ground-truth does not necessarily decrease the likelihood of non-ground-truth subsets.

**Deep Supervised Facility Location.** In this paper, we address the problem of supervised subset selection based on representation learning for a convex relaxation of the uncapacitated facility location function. Facility location is a clustering-based subset selection that finds a set of representatives for which the sum of dissimilarities from every point to the closest representative is minimized [24, 32]. Given the NP-hardness of the problem, different approaches such as convex relaxation [57, 29, 12, 35, 36] and greedy submodular maximization [24, 32] have been proposed to efficiently optimize this utility function. We use convex relaxation because of an appealing property that we exploit: we show conditions under which the sparse convex relaxation recovers ground-truth representatives. We use these conditions to design a loss function to learn representation of data so that inputing each transformed dataset to the facility location leads to finding ground-truth representatives.

Our loss function consists of three terms, a medoid loss that enforces each ground-truth representative be the medoid of its associated cluster, an inter-cluster loss that makes sure there is sufficient margin between points in different clusters induced by ground-truth representatives and an intra-cluster loss that enforces the distances between points in each cluster be smaller than a margin. The latter two loss functions are based on a margin that depends on the regularization parameter of the uncapacitated facility location and the number of points in induced clusters. The conditions and our proposed loss function require knowing the clustering of the data based on assignments to ground-truth representatives. However, computing the assignments requires access to the optimal representation, which is not available. Thus, we propose an optimization scheme that alternates between updating the representation by minimizing our proposed loss given the current assignments of points to ground-truth representatives and updating the assignments given the current representation.

We perform experiments on the problem of supervised instructional video summarization, where each video consists of a set of key-steps (subactivities), needed to achieve a given task. In this case, each training video comes with a list of representative segments/frames, without knowing the labels of representatives and without knowing which representatives across different videos correspond to the same key-step (subactivity), making the supervised subset selection extremely more challenging than classification. Our experiments on two large datasets of ProceL [1] and Breakfast [58] show the effectiveness of our framework.

**Remark 1** *Our setting is different than interactive subset selection [59, 60] that incorporates human supervision* interactively, *i.e., as we run subset selection, we receive and incorporate human feedback to improve subset selection. In our case, we do not have human in the loop interactively. Also, our setting is different than weakly supervised video summarization [61, 62] that use the name of the video categories or additional web data to perform summarization. We assume each dataset has ground-truth summary and do not use additional web data. Finally, [63] uses facility location for metric learning. However, this requires knowledge about assignments of points to predefined categories, which is a stronger requirement than only knowing the ground-truth representatives.*

**Remark 2** *To the best of our knowledge, this is the first work on supervised subset selection that derives conditions for the exactness of a subset selection utility function (i.e., conditions under which subset selection recovers ground-truth representatives) and employs these conditions to design a loss function for representation learning, e.g., via DNNs. In fact, this work takes a major step towards a theoretically motivated supervised subset selection framework.*

**Paper Organization.** The paper is organized as follows. In Section 2, we review the facility location and convex relaxation to solve the subset selection efficiently. In Section 3, we show conditions for the equivalence of the two problems, design a new loss function for representation learning whose minimum satisfies the conditions, hence, guaranteeing to obtain ground-truth representatives, and propose an efficient learning algorithm. In Section 4, we show experimental results on the ProceL and Breakfast datasets for instructional video summarization. Finally, Section 5 concludes the paper.

## 2  Background on Subset Selection

**Facility Location.** Facility location is a clustering-based subset selection utility function, in which each point is assigned to one representative, hence, performing both representative selection and clustering [24]. More specifically, assume we have a dataset $\mathcal{Y} = \{\boldsymbol{y}_1, \ldots, \boldsymbol{y}_N\}$ consisting of $N$ points, for which we are given dissimilarities between pairs of points. Let $d_{i,j} = d(\boldsymbol{y}_i, \boldsymbol{y}_j)$ denote the dissimilarity between points $\boldsymbol{y}_i$ and $\boldsymbol{y}_j$, with $d(\cdot, \cdot)$ being the dissimilarity function. The smaller the $d_{i,j}$ is, the better $\boldsymbol{y}_i$ represents $\boldsymbol{y}_j$. We assume that dissimilarities are non-negative, provide a partial ordering of data and we have $d_{jj} < d_{ij}$ for every $i \neq j$.

In order to find representatives, the facility location selects a subset $\mathcal{S} \subseteq \{1, \ldots, N\}$ of the data points and assigns each point in $\mathcal{Y}$ to the representative point in $\mathcal{S}$ with minimum dissimilarity. In particular, the uncapacitated facility location [64, 65] tries to find a subset $\mathcal{S}$ with a sufficiently small cardinality that gives the best encoding of the dataset, i.e.,

$$\min_{\mathcal{S} \subseteq \{1, \ldots, N\}} \lambda |\mathcal{S}| + \sum_{j=1}^{N} \min_{i \in \mathcal{S}} d_{ij}, \qquad (1)$$

where $\lambda \geq 0$ is a regularization parameter that sets a trade-off between the number of representatives, $|\mathcal{S}|$, and the encoding quality via $\mathcal{S}$. When $\lambda$ is zero, every point will be a representative of itself.

**Sparse Convex Relaxation.** Optimizing the facility location in (1) is NP-hard, as it requires searching over all possible subsets of the dataset. This has motivated efficient algorithms, including forward-backward greedy submodular maximization with worst case performance guarantees [66] as well as sparse convex relaxation [12]. To obtain the convex relaxation, which we use in the paper, one first defines assignment variables $z_{ij} \in \{0, 1\}$, which is 1 when $\boldsymbol{y}_j$ is represented by $\boldsymbol{y}_i$ and is zero otherwise. We can rewrite (1) as an equivalent optimization on the assignment variables as

$$\min_{\{z_{ij}\}} \lambda \sum_{i=1}^{N} \mathrm{I}(\|[z_{i1} \cdots z_{iN}]\|_p) + \sum_{i,j=1}^{N} d_{ij} z_{ij} \quad \text{s.t.} \quad z_{ij} \in \{0, 1\}, \sum_{i=1}^{N} z_{ij} = 1, \ \forall i, j, \qquad (2)$$

where $\mathrm{I}(\cdot)$ is an indicator function, which is one when its argument is nonzero and is zero otherwise. Thus, the first term of the objective function measures the number of representatives, since $[z_{i1} \cdots z_{iN}]$ is nonzero when $\boldsymbol{y}_i$ represents some of the data points and becomes zero otherwise. The second term measures the encoding cost, while the constraints ensure that each point is represented by only one representative.

Notice that (2), which is equivalent to (1), is still an NP-hard problem. Also, (2) is a group-sparse optimization where ideally a few vectors $[z_{i1} \cdots z_{iN}]$ must be nonzero for a few $i$'s that would correspond to the representative points. To obtain an efficient convex relaxation based on group-sparsity (for $p \geq 1$) [12, 29], we drop the indicator function and relaxe the binary constraints to $z_{ij} \in [0, 1]$, hence, solve

$$\min_{\{z_{ij}\}} \lambda \sum_{i=1}^{N} \left\|[z_{i1} \cdots z_{iN}]\right\|_p + \sum_{i,j=1}^{N} d_{ij} z_{ij} \quad \text{s.t.} \quad z_{ij} \geq 0, \ \sum_{i=1}^{N} z_{ij} = 1, \ \forall i, j. \qquad (3)$$

We then obtain the set of representatives $\mathcal{R}$ as points $\boldsymbol{y}_i$ for which $z_{ij}$ is nonzero for some $j$. Moreover, we obtain a clustering of data according to assignments of points to representatives, where for every representative $i \in \mathcal{R}$, we obtain its cluster $\mathcal{G}_i = \{j \in \{1, \ldots, N\} | z_{ij} = 1\}$ as the set of all points assigned to $i$.

# 3 Supervised Facility Location

In this section, we present our proposed approach for supervised subset selection. We discuss conditions under which (3), which is the practical and efficient algorithm for solving the uncapacitated facility location, recovers ground-truth representatives from datasets. We use these conditions to design a loss function for representation learning so that for the transformed data, obtained by minimizing the loss, (3) and equivalently (1) will select ground truth summaries of training datasets. We then present an efficient learning framework to optimize our proposed loss function.

## 3.1 Problem Setting

Assume we have $L$ datasets and their ground-truth representatives, $\{(\mathcal{Y}_\ell, \mathcal{R}_\ell)\}_{\ell=1}^L$, where $\mathcal{Y}_\ell = \{\boldsymbol{y}_{\ell,1}, \dots, \boldsymbol{y}_{\ell,N_\ell}\}$ denotes $N_\ell$ data points in the $\ell$-th dataset and $\mathcal{R}_\ell \subseteq \{1, \dots, N_\ell\}$ denotes the associated set of indices of ground-truth representatives. The goal of supervised subset selection is to train a subset selection method so that the input of each dataset $\mathcal{Y}_\ell$ to the trained model leads to obtaining ground-truth representatives, $\mathcal{R}_\ell$.

In the paper, we fix the subset selection method to the uncapacitated facility location in (1) and consider $p = \infty$ in (2) and (3). We cast the supervised subset selection problem as learning a transformation $f_\Theta(\cdot)$ on the input data so that running the convex algorithm (3) on $f_\Theta(\mathcal{Y}_\ell)$ leads to obtaining $\mathcal{R}_\ell$. We use a deep neural network, parametrized by $\Theta$, for representation learning and use the Euclidean distance as the measure of dissimilarity, i.e., we define

$$d_{i,j}^\ell \triangleq \left\| f_\Theta(\boldsymbol{y}_{\ell,i}) - f_\Theta(\boldsymbol{y}_{\ell,j}) \right\|_2. \tag{4}$$

Notice that we can use other dissimilarities as well (the theory and learning algorithm below work for other dissimilarities), however, the Euclidean distance results in obtaining an embedding space, where points are gathered around ground-truth representatives according to $\ell_2$ distances. To learn the parameters $\Theta$, we design a loss function using conditions that guarantee the performance of (3) for obtaining ground-truth representatives across datasets.

## 3.2 Proposed Learning Framework

We investigate conditions under which the convex algorithm in (3) recovers a given set of points as representatives of transformed data $\{f_\Theta(\boldsymbol{y}_{\ell,1}), \dots, f_\Theta(\boldsymbol{y}_{\ell,N_\ell})\}$. We show that under these conditions, the solution of the convex algorithm in (3), which has the constraint $z_{i,j} \in [0,1]$, will be integer. As a result, the convex relaxation will recover the same solution of the NP-hard non-convex uncapacitated facility location, i.e., the optimality gap between the non-convex and convex formulations vanishes. We then use these conditions to design a loss function for learning the representation parameters $\Theta$.

**Theorem 1** *Consider the convex relaxation of the uncapacitated facility location in* (3)*, with a fixed $\lambda$ and $p = \infty$. Let $\mathcal{R}_\ell$ be the set of ground-truth representatives from the $\ell$-th dataset $\{f_\Theta(\boldsymbol{y}_{\ell,1}), \dots, f_\Theta(\boldsymbol{y}_{\ell,N_\ell})\}$ and let $\mathcal{G}_i^\ell$ denote the cluster associated with the representative $i \in \mathcal{R}_\ell$, i.e.,*

$$\mathcal{G}_i^\ell = \left\{ j \mid i = \operatorname{argmin}_{i'} d_{i',j}^\ell = \operatorname{argmin}_{i'} \| f_\Theta(\boldsymbol{y}_{\ell,i'}) - f_\Theta(\boldsymbol{y}_{\ell,j}) \|_2 \right\}. \tag{5}$$

*The optimization* (3) *recovers $\mathcal{R}_\ell$ as the set of representatives, if the following conditions hold:*

1. *$\forall i \in \mathcal{R}_\ell$, $\forall i' \in \mathcal{G}_i^\ell$, we have $\sum_{j \in \mathcal{G}_i^\ell} d_{i,j}^\ell \le \sum_{j \in \mathcal{G}_i^\ell} d_{i',j}^\ell$;*

2. *$\forall i \in \mathcal{R}_\ell$, $\forall j \in \mathcal{G}_i^\ell$, $\forall i' \notin \mathcal{G}_i^\ell$, we have $\frac{\lambda}{|\mathcal{G}_i^\ell|} + d_{i,j}^\ell < d_{i',j}^\ell$;*

3. *$\forall i \in \mathcal{R}_\ell$, $\forall i', j \in \mathcal{G}_i^\ell$, we have $d_{i',j}^\ell \le \frac{\lambda}{|\mathcal{G}_i^\ell|} + d_{i,j}^\ell$.*

The first condition (medoid condition) states that for points assigned to the cluster of $i \in \mathcal{R}_\ell$, the representative point $i$ must achieve the minimum encoding cost. The second condition (inter-cluster condition) states that the closest point to each cluster from other groups must be sufficiently far from it. The third condition (intra-cluster condition) states that points in the same cluster must not be far from each other. For both the inter and intra cluster conditions, the separation margin is given by $\lambda/|\mathcal{G}_i^\ell|$, depending on the regularization parameter and the number of points in each cluster, i.e., we have an adaptive margin to each cluster. Under the conditions of the Theorem 1, we can show that there is no gap between the NP-hard non-convex formulation in (1) and its convex relaxation in (3).

---

**Algorithm 1 : Supervised Facility Location Learning**

---

**Input:** Datasets $\{\mathcal{Y}_\ell\}_{\ell=1}^L$ and ground truth representatives $\{\mathcal{R}_\ell\}_{\ell=1}^L$.
1: Initialize $\Theta$ by using a pretrained network;
2: **while** (Not Converged) **do**
3:     For fixed $\Theta$, compute $\mathcal{G}_1^\ell, \mathcal{G}_2^\ell, \ldots$ for each dataset $\ell$ via (5);
4:     For fixed $\{\mathcal{G}_1^\ell, \mathcal{G}_2^\ell, \ldots\}_{\ell=1}^L$, update $\Theta$ by minimizing the loss function (7);
5: **end while**

**Output:** Optimal parameters $\Theta$.

---

**Corollary 1** *Under the assumptions of the Theorem 1, the convex relaxation in* (3) *is equivalent to the non-convex uncapacitated facility location optimization in* (1)*, both recovering the same* integer *solution, where for $\mathcal{Y}_\ell$, we recover $\mathcal{R}_\ell$ as the representative set.*

We can also show similar results for $p = 2$ (see the supplementary file). Next, we use the above result to design a loss function for supervised subset selection using the uncapacitated facility location. In fact, if we find a representation $\Theta$ using which the conditions of the Theorem 1 are satisfied, then not only the combinatorial optimization in (1) recovers the ground-truth representatives from each dataset, but also we obtain the same solution using the efficient sparse optimization in (3). To find the desired $\Theta$, we propose a loss function that penalizes violation of the conditions of the Theorem 1. More specifically, we define three loss functions corresponding to the conditions of the theorem, as

$$\mathcal{L}_{medoid}^\ell(\Theta) \triangleq \sum_{i \in \mathcal{R}_\ell} \sum_{i' \in \mathcal{G}_i^\ell} \Big( \sum_{j \in \mathcal{G}_i^\ell} d_{i,j}^\ell - \sum_{j \in \mathcal{G}_i^\ell} d_{i',j}^\ell \Big)_+,$$

$$\mathcal{L}_{inter}^\ell(\Theta) \triangleq \sum_{i \in \mathcal{R}_\ell} \sum_{j \in \mathcal{G}_i^\ell} \sum_{i' \notin \mathcal{G}_i^\ell} \Big( \frac{\lambda}{|\mathcal{G}_i^\ell|} + d_{i,j}^\ell - d_{i',j}^\ell \Big)_+, \qquad (6)$$

$$\mathcal{L}_{intra}^\ell(\Theta) \triangleq \sum_{i \in \mathcal{R}_\ell} \sum_{i',j \in \mathcal{G}_i^\ell} \Big( d_{i',j}^\ell - d_{i,j}^\ell - \frac{\lambda}{|\mathcal{G}_i^\ell|} \Big)_+,$$

where $(x)_+ \triangleq \max\{0, x\}$ is the non-negative thresholding (or ReLU) operator, and $\mathcal{L}_1^\ell, \mathcal{L}_2^\ell, \mathcal{L}_3^\ell$ measure and penalize violation of the medoid, inter-cluster and intra-cluster conditions, respectively, for the dataset $\ell$. Putting the three loss functions together, we propose to minimize the following cost function, defined over the $L$ datasets,

$$\min_\Theta \mathcal{L}(\Theta) \triangleq \sum_{\ell=1}^L \big( \mathcal{L}_{medoid}^\ell(\Theta) + \rho_{inter} \mathcal{L}_{inter}^\ell(\Theta) + \rho_{intra} \mathcal{L}_{intra}^\ell(\Theta) \big), \qquad (7)$$

where $\rho_{inter}, \rho_{intra} \geq 0$ are regularization parameters that set a trade-off between the three terms.

To minimize $\mathcal{L}$, we need to use the clustering of points in every dataset $\mathcal{Y}_\ell$ according to assignments of points to the ground-truth representative set $\mathcal{R}_\ell$, which requires computing $\mathcal{G}_i^\ell$'s. However, computing such clustering via (5) requires knowledge of the optimal representation of the data $\Theta$, which is not available. To address the problem, we propose an efficient learning algorithm that alternates between updating the representation parameters $\Theta$ by minimizing the proposed loss given the current assignments of points to ground-truth representatives and updating the assignments given the current representation. Algorithm 1 shows the steps of our learning algorithm.

Notice that the loss functions naively require considering every representative and every pair of points in the same or different clusters. Given the redundancy of points, this is not often needed and we can only sample a few pairs of points in the same or different clusters to compute each loss.

**Adaptive Margin.** It is important to note that our derived conditions and the loss function make use of a margin $\lambda/|\mathcal{G}_i|$ that depends on the facility location hyperparameter and the number of points in each cluster $\mathcal{G}_i$. In other words, the margin would be different for different clusters during different iterations of our learning scheme. More specifically, for a representative that has few points assigned to it, the size of the cluster would be small, hence, incurring a larger margin than clusters with more number of points. This has the following effect: when a cluster has a small number of points, it could be considered as under-sampled, hence, to generalize better to test data, we need to have a better separation from other clusters, i.e., larger margin. On the other hand, for a cluster with a large number of points, the margin could be smaller as the chance of changing the distances between and within

clusters by adding more samples to it would be low. This is in contrast to contrastive loss functions that use a fixed margin for pairs of dissimilar items, while reducing the distances of similar items as much as possible. Another difference with respect to contrastive loss functions is that in our loss, we compare the encoding quality of each representative point to non-representative points, whereas in contrastive loss, one uses all pairs of similar and dissimilar items.

**Remark 3** *While [35, 36] have shown the integrality of convex relaxation for cardinality-constrained facility location, we showed equivalence conditions for the uncapacitated problem. Moreover, the nature of our conditions, as opposed to asymptotic results, allowed to design the loss in* (7)*. Also, we learn to effectively use a common* $\lambda$ *across different datasets, which cannot be done in the cardinality-constrained case, where the number of ground-truth representatives is already given.*

## 4 Experiments

In this section, we evaluate the performance of our method, which we refer to as Supervised Facility Location (SupFL), as well as other algorithms for learning key-steps (subactivities) of instructional videos by learning from ground-truth summaries. Notice that each training dataset comes with a list of representative segments/frames, without knowing the labels of representatives and without knowing which representatives across different videos correspond to the same key-step (subactivity). This makes the supervised subset selection different and extremely more challenging than classification.

### 4.1 Experimental Setting

**Dataset.** We perform experiments on ProceL [1] and Breakfast [58] datasets. The ProceL is a large multimodal dataset of 12 diverse tasks, such as 'install Chromecast', 'assemble Clarinet', 'perform CPR'. Each task consists of about 60 videos and has a grammar of key-steps, e.g. 'perform CPR' consists of 'call emergency', 'check pulse', 'open airway', 'give compression' and 'give breath'. Each video is annotated with the key-steps. Breakfast is another large dataset of 10 cooking activities by 52 individuals performed in 18 different kitchens. The videos are captured using multiple cameras with different view points. Each activity has approximately 200 videos, corresponding to different views of each person doing the same task, hence a total of 1989 videos in the dataset. Similar to ProceL, each task consists of multiple key-steps (subactivities) required to achieve the task. For example, 'making cereal' consists of 'take a bowl', 'pour cereals', 'pour milk', 'stir cereals', 'sil' (for background frames at the beginning and the end).

For the experiments on ProceL, we split the videos of each task into 70% for training, 15% for validation and 15% for testing. For the Breakfast, we split the videos of each activity into 60% for training, 20% for validation, and 20% for testing. We use the middle segment of each subactivity as the ground-truth representative.

**Feature Extraction and Learning.** Given the similarity of consecutive frames, we perform the subset selection at the segment level. For ProceL, we use the segments provided in the dataset and for Breakfast, we divide each video into segments of 16-frame length with 8 frames overlap between two consecutive segments. We use the C3D network [67] for feature extraction in each segment and use the $4,096$-dimensional feature obtained by the first dense layer after the convolutional layers. We consider two variants of our method: i) SupFL(L), where we learn a linear transformation on the C3D features; ii) SupFL(N), where we learn the parameters of a neural network applied to C3D features. We use Euclidean distance for pairwise dissimilarities.

**Algorithms and Baselines.** We compare the two variants of our method, SupFL(L) and SupFL(N), discussed above, against SubmodMix [52], which learns the weights of a mixture of submodular functions, and dppLSTM[54], which learns to select representatives using a bidirectional LSTM combined with the DPP kernel, and FCSN [68], which learns the weights of a fully convolutional network by treating subset selection as classification of each segment into representative vs non-representative. To show the effectiveness of learning, we also compare with two unsupervised baselines: Uniform, which selects representatives uniformly at random from all segments, and UFL, which corresponds to running the uncapacitated facility location via the forward-backward greedy method on dissimilarities computed via C3D features. This particularly allows to investigate the effectiveness of our method in taking advantage of ground-truth summaries.

**Evaluation metric.** Following [58], we report the segment-wise precision (P), action-wise recall (R) and F1 score (F). These metrics help to measure the performance of finding a representative

| Activity | Uniform | UFL | dppLSTM | SubmodMix | FCSN | SupFL(L) | SupFL(N) |
|---|---|---|---|---|---|---|---|
| perform CPR | 55.7 | 59.7 | 53.4 | 60.0 | 57.4 | 63.7 | **64.9** |
| make coffee | 57.3 | 62.6 | 56.8 | 62.3 | 64.2 | 71.5 | **71.6** |
| jump-start car | 57.2 | 66.0 | 55.8 | 67.2 | 69.6 | 68.5 | **71.4** |
| repot plant | 59.6 | 67.3 | 64.7 | 68.2 | 69.2 | **69.7** | 69.1 |
| change tire | 54.6 | 68.4 | 57.3 | 65.5 | 65.7 | 71.0 | **71.2** |
| tie a tie | 44.6 | 51.6 | 48.1 | 53.5 | **60.2** | 58.5 | 60.0 |
| setup Chromecast | 52.6 | 61.7 | 55.5 | 61.8 | 56.8 | 63.7 | **66.0** |
| change iPhone battery | 53.0 | 55.9 | 53.4 | 61.2 | 59.3 | 62.3 | **63.2** |
| make pbj sandwich | 52.7 | 60.8 | 53.2 | 58.0 | 62.0 | **64.9** | 64.2 |
| make smoke salmon | 59.9 | 69.4 | 62.6 | 71.4 | 65.3 | 72.8 | **74.3** |
| change toilet seat | 55.5 | 61.9 | 56.5 | 62.7 | **68.4** | 66.0 | 67.5 |
| assemble clarinet | 57.8 | 67.2 | 61.7 | 66.0 | 67.8 | **72.0** | 70.5 |
| Average | 55.0 | 62.7 | 56.6 | 63.2 | 63.8 | 67.0 | **67.8** |

Table 1: Average F1 score (%) of different algorithms for subset selection on the ProceL dataset.

for each key-step and the correctness of video segmentation based on assignments of segments to representatives. More specifically, for a video with $N_s$ segments and $N_a$ ground-truth key-steps, after running subset selection we assign each segment to each recovered representative. We compute

$$P = \frac{\hat{N}_s}{N_s}, \;\; R = \frac{\hat{N}_a}{N_a}, \;\; F = \frac{2PR}{P + R}, \tag{8}$$

where $\hat{N}_s$ is the number of the segments that are correctly assigned to representatives, given the ground-truth assignment labels. $\hat{N}_a$ is the number of recovered key-steps in the video via representatives. The F1 score is the harmonic mean between the segment-wise precision and action-wise recall, which is between 0 and 1. We report the average of each score over the videos of each task.

**Implementation details.** We implemented our framework in Pytorch and used the ADMM framework in [12] for subset selection via UFL and our SupFL. We train a model for each individual activity. For SupFL(L), we set the dimension of the transformed data to 1000 and 500 for ProceL and Breakfast, respectively, while for SupFL(N) we set the dimension of the network to $4096 \times 1000 \times 1000$ and $4096 \times 1000 \times 500$ for ProceL and Breakfast, respectively, where we use ReLu activations for the second layer. We use stochastic gradient descent to train our model and use 5 videos in each minibatch. We use the Adam optimizer with the learning rate of 1e-4 and weight decay of 5e-4. We train our model for at most 50 epochs. In order to improve the training time, after we compute assignments of points to each representative in our alternating algorithm, we randomly sample 10 points from each group and use them to form the loss functions in (6). Our method has three hyperparameters $(\lambda, \rho_{inter}, \rho_{intra})$, where $\lambda$ is the regularization of the UFL in (3), while $\rho_{inter}$ and $\rho_{intra}$ are regularization parameters of our loss function in (7). We set the values of hyperparameters using the validation set (we did not perform heavy hyperparameter tuning). In the experiments, we show the effect of the regularization parameters on the performance. To have a fair comparison, we run all methods to select the same number of representatives as the number of ground-truth key-steps in the grammar of the task.

## 4.2 Experimental Results

Table 1 shows the average F1 score (%) of different methods on each task in the ProceL dataset. Notice that our method outperforms other algorithms, obtaining 67.8% and 67.0% F1 score via SupFL(N) and SupFL(L), respectively, over the entire dataset. Compared to the UFL, which is the unsupervised version of our framework, we obtain significant improvement in all tasks, e.g., improving the F1 score by 8.4% and 7.3% for 'tie a tie' and 'change iPhone battery', respectively. dppLSTM, which is supervised, does not do as well as our method and other two supervised algorithms. This comes from the fact that dppLSTM often selects multiple segments from one key-step and from background, due to their appearance diversity, while missing some of the key-steps to choose segments from (see Figure 3). While SubmodMix and FCSN perform better than other baselines, their overall performance is about 4% lower than our method. This comes from the fact that SubmodMix has limited learning capacity, depending on which functions to add, while FCSN treats supervised subset selection as classification, hence embeds ground-truth representative segments (class 1) close to each other and far from non-representative segments (class 0), which is not desired as a representative and a non-representative segment could be very similar.

| Activity | Uniform | UFL | dppLSTM | SubmodMix | SupFL(L) | SupFL(N) |
|---|---|---|---|---|---|---|
| cereals | 58.6 | 63.8 | 58.3 | 64.6 | **66.3** | 63.4 |
| coffee | 73.9 | 77.7 | 78.1 | 79.5 | **82.6** | 80.5 |
| friedegg | 55.2 | 53.8 | **61.2** | 53.4 | 54.9 | 59.7 |
| juice | 61.8 | 67.9 | 65.6 | 67.7 | **72.9** | 71.9 |
| milk | 55.3 | 63.4 | 54.9 | 63.1 | **65.8** | 63.9 |
| pancake | 53.1 | 53.6 | 41.0 | **54.1** | 51.5 | 53.3 |
| salad | 57.5 | 60.5 | 59.3 | 59.4 | **64.5** | 61.2 |
| sandwich | 60.2 | 65.6 | 61.7 | 65.0 | **69.1** | 67.0 |
| scrambledegg | 56.8 | 61.9 | 57.9 | 61.6 | **63.6** | 59.6 |
| tea | 69.2 | 76.8 | 72.6 | 76.1 | **78.1** | 76.3 |
| Average | 60.2 | 64.5 | 61.1 | 64.4 | **66.9** | 65.7 |

Table 2: Average F1 score (%) of different algorithms for subset selection on the Breakfast dataset.

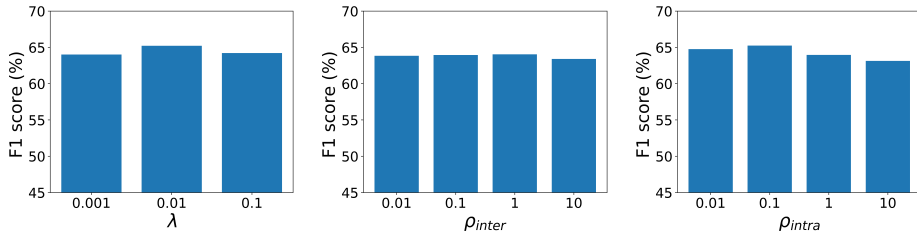

Figure 2: Effect of hyperparameters on the average F1 score (%) over all tasks in the ProceL dataset.

Table 2 shows the average F1 score (%) in the Breakfast dataset[1]. While both versions of our method outperform other algorithms, in contrast to the ProceL, SupFL(L) generally does better than SupFL(N). Moreover, the gap between the performance of UFL and SupFL is smaller. This comes from the fact that the C3D features capture discriminative information for separating different key-steps (subactivities), hence, learning a linear transformation generally does better than a nonlinear one and less improvement will be expected by learning from ground-truth summaries.

Figure 1 shows the average F1 score improvement over not learning data representation on the test videos of the four tasks of 'perform CPR', 'change iPhone battery', 'make coffee' and 'change tire' in ProceL as a function of the number of training epochs. Notice that generally as the training continues the F1 score improves, obtaining between 4% and 10% improvement, depending on the task, over using C3D features.

**Hyperparameter Effect.** We also analyze the performance of our method as a function of the regularization parameters $(\lambda, \rho_{inter}, \rho_{intra})$, where $\lambda$ corresponds to the regularization parameter of the uncapacitated FL utility function in (3), while $\rho_{inter}, \rho_{intra}$ correspond to the hyperparameters that set a trade off between the three terms

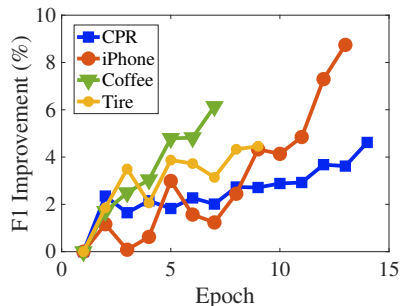

Figure 1: F1 improvement during training on test videos from four tasks in ProceL.

of our loss function in (7). Figure 2 shows the F1 score on the ProceL dataset, where to see the effect of each hyperparameter, we have fixed the values of the other two (these fixed values depend on the task). Notice that the F1 score is relatively stable with respect to the hyperparameter change. In particular, changing $\lambda$ from 0.001 to 0.1 the performance over the dataset changes by at most 1.2% in F1 score, while changing $\rho_{inter}$ and $\rho_{intra}$ from 0.01 to 10, the performance changes by at most 0.6% and 2.1%, respectively.

**Ablation Studies.** To show the effectiveness of using all three loss functions in our proposed cost function in (7), we perform ablation studies. Table 3 shows the average precision, recall and F1 scores on the ProceL dataset. Notice that when we use only one loss or a combination of two loss functions, we achieve relatively similar low scores, being about 7% lower than using the three loss functions together. This shows that, as expected from the theoretical results, we need to use all

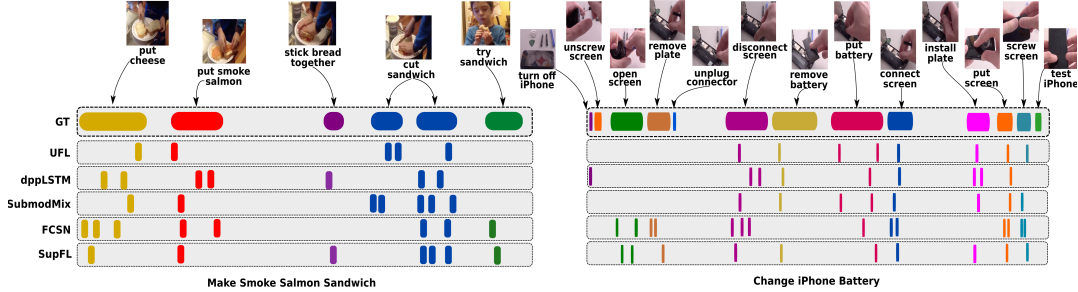

Figure 3: Qualitative results on two test videos from the tasks 'make smoke salmon sandwich' (left) and 'change iPhone battery' (right). Compared to the baselines, our method recovers more number of representatives corresponding to ground-truth key-steps.

| SupFL | Precision | Recall | F1 score |
|---|---|---|---|
| medoid loss | 68.1 | 61.4 | 61.6 |
| inter-cluster loss | 66.2 | 60.2 | 59.9 |
| intra-cluster loss | 67.2 | 57.5 | 59.3 |
| medoid + inter-cluster loss | 68.2 | 60.6 | 61.2 |
| medoid + intra-cluster loss | 68.4 | 60.4 | 61.8 |
| inter-cluster + intra-cluster loss | 64.7 | 57.5 | 58.2 |
| medoid + inter-cluster + intra-cluster loss | **72.8** | **66.3** | **67.8** |

Table 3: Average performance our method, SupFL(N), on ProceL with different combinations of loss functions.

loss functions corresponding to the three theoretical conditions in order to effectively learn from ground-truth summaries. Also, notice that the medoid loss alone or its combination with either of the two other losses obtains slightly better performance than using the inter-cluster or intra cluster loss or their combination. This is expected as the medoid loss tries to center points around each ground-truth representative. Finally, the combination of the inter-cluster and intra-cluster loss, which has weak resemblance to the contrastive loss, does not do well in the supervised subset selection problem.

**Qualitative Results.** Figure 3 shows a qualitative result of running different methods for two videos from the two tasks of 'change iPhone battery' and 'make smoke salmon sandwich' from the ProceL dataset, where all methods choose the same number of representatives (for clarity, we do not show representatives obtained from background). Notice that for 'smoke salmon sandwich' our method correctly finds representatives from all key-steps, while other methods miss one of the key-steps. Similarly, for 'change iPhone screen', our method is more successful than baselines, which miss 5 or 6 key-steps. Our method in general does better in obtaining diverse representative segments, while other supervised baselines often obtain multiple redundant representatives from the same key-step.

## 5 Conclusions

We addressed the problem of supervised subset selection by generalizing the facility location to learn from ground-truth summaries. We considered an efficient sparse optimization of the uncapacitated facility location and investigated conditions under which it recovers ground-truth representatives and also becomes equivalent to the original NP-hard problem. We designed a loss function and an efficient framework to learn representations of data so that the input of transformed data to the facility location satisfies the theoretical conditions, hence, recovers ground-truth summaries. We showed the effectiveness of our method for recovering key-steps of instructional videos. To the best of our knowledge, this is the first work on supervised subset selection that derives conditions under which subset selection recovers ground-truth representatives and employs them to design a loss function for deep representation learning. We believe that this work took a major step towards a theoretically motivated supervised subset selection framework.

## Acknowledgements

This work is supported by DARPA Young Faculty Award (D18AP00050), NSF (IIS-1657197), ONR (N000141812132) and ARO (W911NF1810300). Chengguang Xu would like to thank Dat Huynh and Zwe Naing for their help and advice with some of the implementations during his research assistantship at MCADS lab, which resulted in this work.

## Footnotes

[1]FCSN on Breakfast produced significantly lower F1 scores compared to all other baselines.

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
