[Supplementary Material · neurips19_supSS_supmat_final.pdf]

# Supplementary Materials
# Deep Supervised Summarization:
# Algorithm and Application to Learning Instructions

**Chengguang Xu**
Khoury College of Computer Sciences
Northeastern University
Boston, MA 02115
xu.cheng@husky.neu.edu

**Ehsan Elhamifar**
Khoury College of Computer Sciences
Northeastern University
Boston, MA 02115
eelhami@ccs.neu.edu

## 1   Proof of Theoretical Results

We prove the result of Theorem 1 in the main paper. Recall that the uncapacitated facility location optimization can be written as

$$\min_{\{z_{ij}\}} \lambda \sum_{i=1}^{N} \mathrm{I}(\|[z_{i1} \cdots z_{iN}]\|_\infty) + \sum_{i,j=1}^{N} d_{ij} z_{ij} \quad \text{s.t.} \quad z_{ij} \in \{0,1\}, \sum_{i=1}^{N} z_{ij} = 1, \ \forall i,j, \qquad (1)$$

where, given the binary assumption on $z_{ij}$, we have $\mathrm{I}(\|[z_{i1} \cdots z_{iN}]\|_\infty) = \|[z_{i1} \cdots z_{iN}]\|_\infty$. The convex relaxation of the problem is given by

$$\min_{\{z_{ij}\}} \lambda \sum_{i=1}^{N} \|[z_{i1} \cdots z_{iN}]\|_\infty + \sum_{i,j=1}^{N} d_{ij} z_{ij} \quad \text{s.t.} \quad z_{ij} \geq 0, \ \sum_{i=1}^{N} z_{ij} = 1, \ \forall i,j. \qquad (2)$$

We restate the theorem, where for simplicity of notation, we remove the super or subscript $\ell$, corresponding to the $\ell$-th dataset, and consider a generic dataset $\mathcal{Y}$ with $\mathcal{R}$ being its set of representatives.

**Theorem 1** *Consider the convex relaxation of the uncapacitated facility location in* (2)*, with a fixed $\lambda$ and $p = \infty$. Let $\mathcal{R}$ be the set of ground-truth representatives from a dataset $\{f_\Theta(\boldsymbol{y}_1), \ldots, f_\Theta(\boldsymbol{y}_N)\}$ and let $\mathcal{G}_i$ denote the cluster associated with the representative $i \in \mathcal{R}$, i.e.,*

$$\mathcal{G}_i = \{j \mid i = \mathrm{argmin}_{i'} d_{i',j} = \mathrm{argmin}_{i'} \|f_\Theta(\boldsymbol{y}_i) - f_\Theta(\boldsymbol{y}_j)\|_2\}. \qquad (3)$$

*The optimization* (2) *recovers $\mathcal{R}$ as the set of representatives, if the following conditions hold:*

1. *Medoid condition: $\forall i \in \mathcal{R}$, $\forall i' \in \mathcal{G}_i$, we have $\sum_{j \in \mathcal{G}_i} d_{i,j} \leq \sum_{j \in \mathcal{G}_i} d_{i',j}$;*

2. *Inter-cluster condition: $\forall i \in \mathcal{R}$, $\forall j \in \mathcal{G}_i$, $\forall i' \notin \mathcal{G}_i$, we have $\frac{\lambda}{|\mathcal{G}_i|} + d_{i,j} < d_{i',j}$;*

3. *Intra-cluster condition: $\forall i \in \mathcal{R}$, $\forall i', j \in \mathcal{G}_i$, we have $d_{i',j} \leq \frac{\lambda}{|\mathcal{G}_i|} + d_{i,j}$.*

*Proof.* First, we convert the optimization in (2) into an equivalent linear program, by introducing auxiliary variables $\zeta_i$ and solving

$$\min_{\{z_{ij}\},\{\zeta_i\}} \sum_{i,j=1}^{N} d_{ij} z_{ij} + \lambda \sum_{i=1}^{N} \zeta_i \quad \text{s.t.} \quad z_{ij} \geq 0, \ z_{ij} \leq \zeta_i, \ \forall i, j; \ \sum_{i=1}^{N} z_{ij} = 1, \ \forall j. \qquad (4)$$

We form the Lagrangian function $\mathcal{L}$, by introducing Lagrange multiplier $\gamma_{ij} \geq 0$ associated with $z_{ij} \geq 0$, $\alpha_{ij} \geq 0$ associated with $z_{ij} \leq \zeta_i$ and $\theta_j \in \mathbb{R}$ associated with $\sum_{i=1}^{N} z_{ij} = 1$. The Lagrangian

function is given by

$$\mathcal{L} = \sum_{i=1}^{N}\sum_{j=1}^{N} d_{ij}z_{ij} + \lambda\sum_{i=1}^{N}\zeta_i + \sum_{j=1}^{N}\theta_j\Big(1 - \sum_{i=1}^{N} z_{ij}\Big) - \sum_{i=1}^{N}\sum_{j=1}^{N}\gamma_{ij}z_{ij} + \sum_{i=1}^{N}\sum_{j=1}^{N}\alpha_{ij}(z_{ij} - \zeta_i). \quad (5)$$

The Karush-Kuhn-Tucker (KKT) optimality conditions (stationarity, primal feasibility, dual feasibility and complementary slackness) for (4) are given by

$$S_1: \quad \frac{\partial\mathcal{L}}{\partial z_{ij}} = d_{ij} - \theta_j - \gamma_{ij} + \alpha_{ij} = 0, \quad (6)$$

$$S_2: \quad \frac{\partial\mathcal{L}}{\partial\zeta_i} = \lambda - \sum_{j=1}^{N}\alpha_{ij} = 0, \quad (7)$$

$$P_1: \quad z_{ij} \geq 0, \quad (8)$$

$$P_2: \quad \sum_{i=1}^{N} z_{ij} = 1, \quad (9)$$

$$P_3: \quad z_{ij} \leq \zeta_i, \quad (10)$$
$$D_1: \quad \gamma_{ij} \geq 0, \quad (11)$$
$$D_2: \quad \alpha_{ij} \geq 0, \quad (12)$$
$$C_1: \quad \alpha_{ij}(z_{ij} - \zeta_i) = 0, \quad (13)$$
$$C_2: \quad \gamma_{ij}z_{ij} = 0. \quad (14)$$

Given that in (4), the objective function is convex, the inequality constraints are continuously differentiable, and the equality constraints are affine, the KKT conditions are necessary and sufficient for the optimality. Thus, if we find $\big(\{z_{ij}\},\{\zeta_i\},\{\alpha_{i,j}\},\{\gamma_{ij}\},\{\theta_j\}\big)$ satisfying KKT conditions, $\big(\{z_{ij}\},\{\zeta_i\}\big)$ would be the optimal solution of (4), hence $\{z_{ij}\}$ would be the solution of (2).

Next, we show that under the conditions of the theorem, $\mathcal{R}$ would be the set of representatives and each point will be only assigned to the closest representative (assignment variable being 1) according to the dissimilarity values. Let

$$\mathcal{G}_i \triangleq \big\{ j \in \{1, \ldots, N\} \mid i = \operatorname{argmin}_{i' \in \mathcal{R}} d_{i',j} \big\}, \quad (15)$$

denote the set of points closets to the representative $i$ in $\mathcal{R}$. Also, let $M(j)$ denote the index of the closest ground-truth representative to the point $j$, i.e.,

$$M(j) \triangleq \operatorname{argmin}_{i' \in \mathcal{R}} d_{i',j}. \quad (16)$$

We want to show that, under the conditions of the theorem, we have

$$\begin{aligned} &a)\ z_{ij}^* = 1, \zeta_i^* = 1,\ \forall i \in \mathcal{R}, \forall j \in \mathcal{G}_i, \\ &b)\ z_{ij}^* = 0, \zeta_i^* = 1,\ \forall i \in \mathcal{R}, \forall j \notin \mathcal{G}_i, \\ &c)\ z_{ij}^* = 0, \zeta_i^* = 0,\ \forall i \notin \mathcal{R}, \forall j. \end{aligned} \quad (17)$$

Notice that, given the fact $z_{ij} \leq \zeta_i$ for all $j$, and the fact the we minimize $\sum_i \zeta_i$ in the objective function, we must have $\zeta_i = 1$, whenever $i$ is a representative point and $\zeta_i = 0$, whenever $i$ is not a representative point.

To prove that (17) holds under the assumptions of the theorem, we introduce dual certificates for which KKT conditions are satisfied. More specifically, we let

$$\begin{aligned} &d)\ \alpha_{i,j}^* = \frac{\lambda}{|\mathcal{G}_i|},\ \gamma_{ij}^* = 0,\ \forall i \in \mathcal{R}, \forall j \in \mathcal{G}_i, \\ &e)\ \alpha_{i,j}^* = 0,\ \gamma_{ij}^* = d_{ij} - \theta_j^*,\ \forall i \in \mathcal{R}, \forall j \notin \mathcal{G}_i, \\ &f)\ \alpha_{i,j}^* \geq (\theta_j^* - d_{ij})_+,\ \gamma_{ij}^* = (\theta_j^* - d_{ij})_+ - (\theta_j^* - d_{ij}),\ \forall i \notin \mathcal{R}, \forall j, \\ &g)\ \theta_j^* = \frac{\lambda}{|\mathcal{G}_i|} + d_{M(j)j},\ \forall j. \end{aligned} \quad (18)$$

It is easy to verify that a) to c) satisfy primal feasibility conditions ($P_1 - P_3$), d), f) and g) satisfy dual feasibility conditions ($D_1 - D_2$), e) satisfies $D_2$, the stationarity condition $S_1$ and complementary slackness ($C_1 - C_2$) are always satisfied, and the stationarity condition $S_2$ is always satisfied for d) and e).

We need to show that $\gamma_{ij}^*$ in e) is dual feasible, and that $\alpha_{ij}^*$ in f) satisfies $S_2$. Given the choice of $\theta_j^*$ in g), we have

$$\gamma_{ij}^* = d_{ij} - \theta_j^* = d_{ij} - \frac{\lambda}{|\mathcal{G}_i|} - d_{M(j)j} \geq 0, \tag{19}$$

where the non-negativity holds using the intra-cluster condition. Also, we want to show that $\forall i \notin \mathcal{R}$, for the $\alpha_{ij}^*$ in f) we have $S_2$ being satisfied. Notice that, given $\theta_j^*$ in g), we can write

$$\sum_{j=1}^N (\theta_j^* - d_{ij})_+ = \sum_{j:M(j)=M(i)} (\theta_j^* - d_{ij})_+ + \sum_{j:M(j)\neq M(i)} (\theta_j^* - d_{ij})_+$$

$$= \sum_{j:M(j)=M(i)} (\frac{\lambda}{|\mathcal{G}_{M(j)}|} + d_{M(j)j} - d_{ij})_+ + \sum_{j:M(j)\neq M(i)} (\frac{\lambda}{|\mathcal{G}_{M(j)}|} + d_{M(j)j} - d_{ij})_+. \tag{20}$$

From the inter-cluster condition of the theorem, we have

$$(\frac{\lambda}{|\mathcal{G}_{M(j)}|} + d_{M(j)j} - d_{ij})_+ = \frac{\lambda}{|\mathcal{G}_{M(j)}|} + d_{M(j)j} - d_{ij}, \ \forall j : M(j) = M(i), \tag{21}$$

while from the intra-cluster condition of the theorem, we have

$$(\frac{\lambda}{|\mathcal{G}_{M(j)}|} + d_{M(j)j} - d_{ij})_+ = 0, \ \forall j : M(j) \neq M(i). \tag{22}$$

Thus, we can rewrite (20) as

$$\sum_{j=1}^N (\theta_j^* - d_{ij})_+ = \sum_{j:M(j)=M(i)} (\frac{\lambda}{|\mathcal{G}_{M(j)}|} + d_{M(j)j} - d_{ij}) = \lambda + \sum_{j:M(j)=M(i)} (d_{M(j)j} - d_{ij}). \tag{23}$$

Notice that from the medoid condition of the theorem, we have $\sum_{j:M(j)=M(i)} d_{M(j)j} \leq \sum_{j:M(j)=M(i)} d_{ij}$, hence,

$$\sum_{j=1}^N (\theta_j^* - d_{ij})_+ \leq \lambda. \tag{24}$$

As a result, by appropriately choosing $\alpha_{ij}$ as in f), the stationarity condition $S_2$ will be satisfied. This completes the proof.

Notice that, we have also shown that the solution of the convex optimization will be integer, under the assumptions of the theorem, where each point is assigned to its representative with $z_{ij} = 1$. This, in addition to the fact that the optimal value of the convex optimization is a lower bound for the non-convex optimization (the integer constrained is relaxed), implies the following: under the conditions of the theorem, the gap between the convex and non-convex optimization vanishes and the two problems would be equivalent.

∎

Below, we show a similar result for $p = 2$.

**Theorem 2** *Consider the convex relaxation of the uncapacitated facility location in (2), with a fixed $\lambda$ and $p = 2$. Let $\mathcal{R}$ be the set of ground-truth representatives from a dataset $\{f_\Theta(\boldsymbol{y}_1), \ldots, f_\Theta(\boldsymbol{y}_N)\}$ and let $\mathcal{G}_i$ denote the cluster associated with the representative $i \in \mathcal{R}$, i.e.,*

$$\mathcal{G}_i = \{j \mid i = \operatorname{argmin}_{i'} d_{i',j} = \operatorname{argmin}_{i'} \|f_\Theta(\boldsymbol{y}_i) - f_\Theta(\boldsymbol{y}_j)\|_2\}. \tag{25}$$

*The optimization (2) recovers $\mathcal{R}$ as the set of representatives, if the following conditions hold:*

    *1. $\forall i \in \mathcal{R}$, $\forall i' \in \mathcal{G}_i$, we have $\sum_{j \in \mathcal{G}_i} d_{i,j} \leq \sum_{j \in \mathcal{G}_i} d_{i',j}$;*

2. $\forall i \in \mathcal{R}$, $\forall j \in \mathcal{G}_i$, $\forall i' \notin \mathcal{G}_i$, we have $\frac{\lambda}{\sqrt{|\mathcal{G}_i|}} + d_{i,j} < d_{i',j}$;

3. $\forall i \in \mathcal{R}$, $\forall i', j \in \mathcal{G}_i$, we have $d_{i',j} \leq \frac{\lambda}{\sqrt{|\mathcal{G}_i|}} + d_{i,j}$.

4. $\forall i \in \mathcal{R}$, $\forall i', j \in \mathcal{G}_i$, we have $d_{i,j} \leq (1 - \frac{1}{\sqrt{|\mathcal{G}_i|}})\lambda + d_{i',j}$.

Notice that in this case, we need an additional intra-cluster condition that makes sure that the radius of each cluster is sufficiently small. Notice that compared to the case of $p = \infty$, the margin for $p = 2$ has changed to $\lambda/\sqrt{|\mathcal{G}_i|}$. Given the additional constraint and the fact that designing a loss function for the case of $p = 2$ requires four terms, as opposed to three with $p = \infty$, we chose to use $p = \infty$ in the paper. We do not show the proof as we do not use it in the paper, however, the derivations are similar to $p = \infty$.