[Reviews · NeurIPS 2019]

Reviewer 1



The paper is well-written and easy to follow. The main contributions are clear and justified, and they follow quite directly from the concept of using facility location and clustering to do learning. The technical ideas are strongly related to clustering, from the clustering-based loss functions to the EM/K-means flavor of the main algorithm. The experimental results quite thoroughly demonstrate the value of the proposed method on a dataset, though perhaps more datasets could be used. [Update after Author Feedback]. I thank the authors for their detailed responses to the reviewer questions.

Reviewer 2



This paper proposes a sparse convex relation of the facility location utility function for subset selection, for the problem of recovering ground-truth representatives for datasets. This relaxation is used to develop a supervised learning approach for this problem, which involves a learning algorithm that alternatively updates three loss functions (Eq. 7 and Alg. 1) based on three conditions for which this relaxation recovers ground-truth representatives (Theorem 1). The supervised facility learning approach described in this paper appears to be novel, and is described clearly. The experimental results are reasonably convincing overall. One weakness is that only one dataset is used, the Breakfast dataset. It would be more convincing to include results for at least one other dataset. The epoch 0 (before training) TSNE visualization is unnecessary and can be removed. Also, the interpretation of Fig. 3 provided in the paper somewhat subjective and not completely convincing. For example, while the authors point out that the SubModMix and dppLSTM baselines can select multiple representatives from the same activity, SupUFL-L (one of the proposed approaches in the paper) can also select multiple representatives from the same activity. Also, while the baselines fail to select representatives from the “take cup” subactivity, SupUFL-L fails to select representatives from the SIL subactivity. Finally, error bars (confidence estimates) should be provided for the scores in Tables 1 and 2.

Reviewer 3



Minor comment: -Probably a naive comment -- In (7), there may be some trivial \Theta, (say \Theta=0 in some settings) which enforce all f_\Theta(y) to be equal so that all the data points will be assigned to one cluster. In practice, a random initialization of \Theta and an iterative gradient algorithm may result in a reasonably good \Theta, but the problem in (7) is not what is being solved? -In experiments, is it possible to include a tweaked version of [59] as another baseline?

[Author Response · NeurIPS 2019]

We would like to thank the reviewers for their feedback and their encouraging comments about the theoretical
contribution [R1, R2, R3], novelty of the learning approach [R1, R2], convincing experiments [R1, R2, R3], and the
paper being well-written [R1, R2]. Our responses are as follows.

**Response to Reviewer 1:**

**C1:** "The experimental results quite thoroughly demonstrate the value of the proposed method on a dataset, though
perhaps more datasets could be used"

**Answer:** Thanks. To address this comment, we performed experiments on the Inria instructional video dataset (5 tasks
and 30 videos per task). We used C3D features of video segments and ran all subset selection methods with the same
settings discussed in the paper. The results in Table 1 (below) show significant improvement w.r.t. the state of the art.

**C2:** "If I have misjudged the novelty of the work, please explain why."

**– Answer:** We would like to mention that this is the first work on supervised subset selection that derives conditions for
the exactness of a subset selection utility function (i.e., conditions under which subset selection recovers ground-truth
representatives) and employs these conditions to design a loss function for representation learning, e.g., via DNNs. In
fact, our work takes a major step towards a theoretically correct/motivated supervised subset selection framework. We
also plan to release the code, if accepted.

We hope that our responses address the reviewer's comments and kindly ask the reviewer to raise his/her rating.

**Response to Reviewer 2:**

**C1:** "It would be more convincing to include results for at least one other dataset"

**Answer:** Thanks. To address this, we performed experiments on the Inria instructional video dataset, which has 5 tasks
and 30 videos per task. We used C3D features of video segments and ran all methods with the same settings discussed
in the paper. Table 1 (below) shows that our method outperforms other algorithms with at least $5\%$ on the entire dataset.

**C2:** "error bars (confidence estimates) should be provided for the scores in Tables 1 and 2."

**– Answer:** Thanks for the comment. We will update the tables to include score variances. We would like to mention
that the variance of our method (SupUFL-L and SupUFL-L) for each task is less than $0.7\%$ in Table 1 in the main paper.

| Activity | Uniform | UFL | dppLSTM | SubmodMix | SupUFL-L | SupUFL-N |
|---|---|---|---|---|---|---|
| cpr | 51.6 | 55.4 | 59.8 | 61.6 | 65.3 | **66.4** |
| coffee | 53.4 | 52.2 | 47.3 | 56.1 | 58.4 | **61.3** |
| repot | 50.8 | 56.5 | 67.2 | 60.8 | **74.8** | 74.3 |
| jump car | 52.7 | 55.9 | 57.4 | 57.5 | 62.8 | **66.7** |
| changing tire | 50.1 | 59.3 | 55.3 | 62.8 | 65.4 | **69.2** |
| Average | 51.7 | 55.9 | 57.4 | 59.8 | 65.3 | **67.6** |

Table 1: F1 score (%) of different algorithms on the Inria dataset.

**Response to Reviewer 3:**

**C1:** "This tightness result is interesting but not novel"

**Answer:** While tightness of convex relaxation for some non-convex problems has been studied before, this is the first
work studying the exactness for the uncapacitated facility location function. Moreover, we use our derived exactness
conditions to design a loss function for representation learning in the supervised setting, hence, moving towards a
theoretically correct/motivated supervised subset selection.

**C2:** "In experiments, is it possible to include a tweaked version of [59] as another baseline?"

**Answer:** Thanks for the question. The reason we cannot simply use [59] for supervised subset selection is that not
only it requires knowing assignment of points to predefined categories (please see Remark 1), but also we need to
know matching between ground-truth representatives of different videos, i.e., we need to know which ground-truth
representative in each video belong to the same subactivity/category. Thus, to extend [59] to supervised subset selection
we need to iteratively i) perform matching/clustering of ground-truth representatives across all videos, ii) assign
segments to induced categories, iii) perform learning via [59]. In our initial experiments, this did not do well due to the
difficulty of matching ground-truth representatives across videos (also not all videos have all subactivities). We will
explain this further in the revised paper, if accepted.

**C3:** "In (7), there may be some trivial $\Theta$, (say $\Theta = 0$ in some settings) which enforce all $f_\Theta(y)$ to be equal so that
all the data points will be assigned to one cluster. In practice, a random initialization of $\Theta$ and an iterative gradient
algorithm may result in a reasonably good $\Theta$, but the problem in (7) is not what is being solved?"

**Answer:** Thanks for the question. In our case, each ground-truth representative of a dataset form its own group and
our third loss function, $\mathcal{L}_3^\ell(\Theta)$ (equation (6)), enforces that different representatives must be separated by the adaptive
margin. This prevents data to collapse to the same point (for which the total loss function is positive, whereas when all
our three conditions are satisfied, the total loss will be zero).

[Meta-Review · NeurIPS 2019]

The paper presents a supervised facility location based approach to subset selection, i.e., choosing a set of representative points from a new dataset. The paper considers a sparse convex relaxation of the problem and characterizes conditions for getting integral solutions. An alternating algorithm utilizing the integral solutions is presented for learning the subset mapping. Extensive experimental results are presented to illustrate the effectiveness of the proposed approach. The reviewers agree that the paper makes a novel contribution to an important problem and the paper is well written.